

# Survival probability in Generalized Rosenzweig-Porter random matrix ensemble

Giuseppe De Tomasi[1,2*] Mohsen Amini[3], Soumya Bera[4],
Ivan M. Khaymovich[1] and Vladimir E. Kravtsov[5,6,7]

**1** Max-Planck-Institut für Physik komplexer Systeme,
Nöthnitzer Straße 38, 01187 Dresden, Germany
**2** Technische Universität München, 85747 Garching, Germany
**3** Department of Physics, University of Isfahan(UI) - Hezar Jerib, 81746-73441, Isfahan, Iran
**4** Department of Physics, Indian Institute of Technology Bombay, Mumbai 400076, India
**5** Abdus Salam International Center for Theoretical Physics,
Strada Costiera 11, 34151 Trieste, Italy
**6** L. D. Landau Institute for Theoretical Physics - Chernogolovka, Russia
**7** Kavli Institute for Theoretical Physics, Kohn Hall, University of California,
Santa Barbara, CA 93106-4030, U.S.A.

⋆ detomasi@pks.mpg.de

## Abstract

We study analytically and numerically the dynamics of the generalized Rosenzweig-Porter model, which is known to possess three distinct phases: ergodic, multifractal and localized phases. Our focus is on the survival probability $R(t)$, the probability of finding the initial state after time $t$. In particular, if the system is initially prepared in a highly-excited non-stationary state (wave packet) confined in space and containing a fixed fraction of all eigenstates, we show that $R(t)$ can be used as a dynamical indicator to distinguish these three phases. Three main aspects are identified in different phases. The ergodic phase is characterized by the standard power-law decay of $R(t)$ with periodic oscillations in time, surviving in the thermodynamic limit, with frequency equals to the energy bandwidth of the wave packet. In multifractal extended phase the survival probability shows an exponential decay but the decay rate vanishes in the thermodynamic limit in a non-trivial manner determined by the fractal dimension of wave functions. Localized phase is characterized by the saturation value of $R(t \to \infty) = k$, finite in the thermodynamic limit $N \to \infty$, which approaches $k = R(t \to 0)$ in this limit.

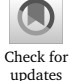
## 1  Introduction

Existence of a transition between a diffusive metal and a perfect insulator at a finite energy density in disordered *interacting* quantum systems isolated from the environment [1–3] has attracted a lot of attention and generated a rapidly expanding field of Many-Body Localization (MBL). However, the expected diffusive behavior of a normal metal in the delocalized phase of such systems has not been completely confirmed in several numerical simulations. Instead a sub-diffusive behavior has been often found [4–11] raising a question of an existence of the non-ergodic extended (NEE) phase. Indeed, in such a phase, often nicknamed as "bad metal" [16, 17], many-body wave functions are neither localized nor occupying all the available Hilbert space, but are like multifractal one-particle wave functions at the transition point of an ordinary Anderson localization problem [18]. The existence of NEE phase is of principle importance as it implies breakdown of conventional Boltzmann statistics. Moreover, the hierarchical, multifractal structure of eigenstates and hence of local spectrum (fractal mini-bands [17]) in interacting qubit systems is important for a rapidly developing part of computer science, the *machine learning* [19], as it allows for a continuous learning process with focusing on progressively more detailed properties of a learning subject. Nevertheless, the persistence of this NEE phase in the thermodynamic limit is still under debate, specifically, in recent works it has been argued that the NEE phase might only be transient, which implies that it crossover to a diffusive phase at long times in the thermodynamic limit [20, 21]. For this reason it is of fundamental importance to study a model in which this phase is rigorously proved to exist in order to give an efficient criteria to characterize it in generic cases.

One of the consequences of the lack of ergodicity in such a phase is the behavior of the survival probability $R(t) = |\langle \Psi(t)|\Psi(0)\rangle|^2$ at large times $t \to \infty$ in a system of finite dimension $N$ of the Hilbert space (see, e.g., the discussion of $R(t)$ [12] in the model suggested in the

seminal work [13],which laid foundation to the modern developments in the field of MBL by discovering a mapping between the interacting disordered many-body systems and non-interacting Anderson localization in the Fock space), with $|\Psi(0)\rangle$ being the initially prepared state,

$$R(t \to \infty) \propto N^{-D} . \tag{1}$$

Here $D$ is the eigenfunction fractal dimension. Ergodicity implies an asymptotic equipartition over all available many-body states, and thus $D = 1$, while in an MBL phase only a finite number of states are involved, and therefore $D = 0$ [1]. The NEE phase is characterized by $0 < D < 1$, so that the volume in the Hilbert space occupied at $t \to \infty$ is infinite $N^D \to \infty$ in the thermodynamic limit, yet it constitutes zero fraction $N^D/N \to 0$ of all states. The survival probability $R(t)$ is of particular importance for many-body spin systems, as it is a proxy of the spin autocorrelation function $\langle \sigma(0)\sigma(t) \rangle$ in interacting spin-qubit systems [22].

A simple *single-particle* system where all three phases, ergodic, localized and NEE, exist in the corresponding range of parameters, is the Generalized Rosenzweig-Porter (GRP) random matrix ensemble [23, 24] [2]. The GRP is not only the simplest toy model for the transition into the NEE phase. It has been shown very recently [25] that the simplest model of quantum spin glass, the Quantum Random Energy Model (QREM), reduces to GRP model and the spin autocorrelation function in QREM shows the typical features of the survival probability of GRP ensemble that we are studying in this paper. Even more recently it has been shown that the SYK-model [26,27] perturbed by a single-body term playing a role of the diagonal disorder is in the same universality class as GRP model [28–30]. The latter works have conjectured the presence of a whole NEE phase in the perturbed SYK-model in the parametrically wide range of the disorder strengths.

The GRP ensemble [24] of random real matrices is characterized by independent identically distributed entries $H_{nm}$, such that:

$$\overline{H_{nm}} = 0, \quad \overline{(H_{nn})^2} = 1, \quad \overline{(H_{n \neq m})^2} = \frac{\lambda^2}{N^\gamma}, \tag{2}$$

where $\overline{A}$ denotes ensemble average of $A$, $N$ is the matrix size, and $\gamma$, $\lambda$ are real parameters. The existence of the NEE phase and a transition to the ergodic state in this model has been recently suggested in Ref. [24], and further confirmed in Ref. [34–36], culminating with a rigorous proof in Ref. [37]. In particular, at $\gamma < 1$ the system is fully ergodic ($D = 1$) and behaves as the Gaussian ensemble [38], at $1 < \gamma < 2$ the NEE phase is realized with $D = 2 - \gamma$ and for $\gamma > 2$ the system is localized ($D = 0$). The critical points

$$\gamma_E = 1, \quad \gamma_c = 2 \tag{3}$$

indicate the transitions from the NEE phase to ergodic (*ergodic* transition) and localized phases (*localization* transition), respectively. In this paper we consider the time dependence of the survival probability

$$R(t) = \overline{\left| \langle \Psi_0 | \widehat{P}_f e^{-it\widehat{H}} \widehat{P}_f | \Psi_0 \rangle \right|^2}, \tag{4}$$

of the initial state $|\Psi(0)\rangle = \widehat{P}_f |\Psi_0\rangle$, being a single-particle analogue of a product state $|\Psi_0\rangle$ represented by the basis vector in which only one component is 1 and all the others are equal to 0, projected by $\widehat{P}_f = \sum_{n=N/2(1-f)}^{N/2(1+f)} |\psi_n\rangle\langle\psi_n|$ to a finite small fraction $0 < f \ll 1$ of eigenstates $\{|\psi_n\rangle\}$ of $\widehat{H}$ in the middle of the spectrum. The final state $\Psi(t \to \infty)$ is a random $N$-vector in

---

[1]Although there are works claiming finite $D > 0$ in MBL phase (see, e.g., [14,15]), the value of $D$ is quite small, nearly within the error bar.

[2]Recently, the presence of a whole NEE phase has been found in several static [31, 32] and periodically-driven [33] long-range random matrix models from a different universality class.

which $N^D$ elements are of the order of $\sim N^{-D}$ and all the others are much smaller. The main reason to use the projector of the considered form is that such $\widehat{P}_f$ provides an opportunity to use $R(t)$ as a sensitive dynamical indicator of the ergodic phase. On the other hand, it is helpful for numerical reasons, as it reduces finite size effects. Indeed, usually the finite size corrections to the fractal dimension depend on the energy ($D = D(E, N)$), thus projecting to a small fraction of eigenstates $f \ll 1$ one can neglect these variations in $D$ versus $N$ ($D(E, N) \approx D(0, N)$), making the data analysis easier.

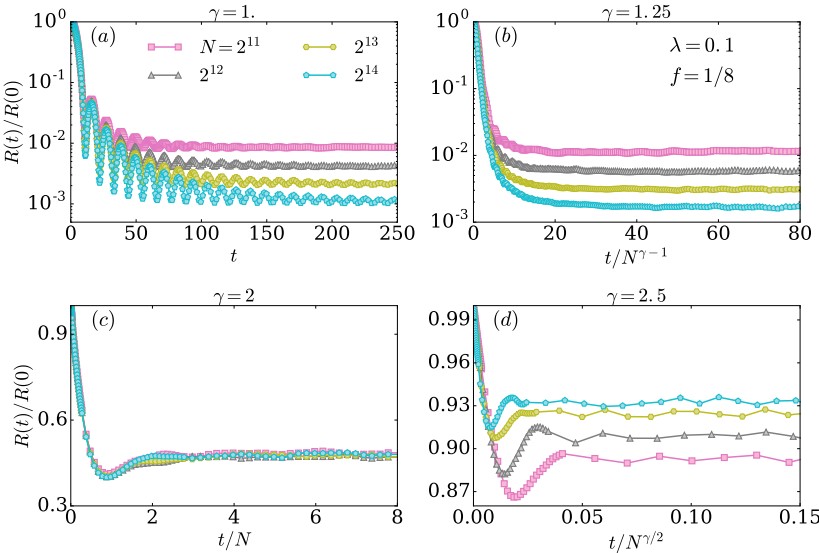

Figure 1: (Color online) Evolution of $R(t)$ with increasing $\gamma$ for the box distribution of on-site disorder. The ergodic transition is marked by the onset of oscillations in $R(t)$ which survive the thermodynamic limit $N \to \infty$. The value of $R(\infty) \propto N^{-D}$ decreases with increasing the system size for $\gamma < 2$, while for $\gamma \geq 2$ it increases and approaches its initial value $R(0)$ in the thermodynamic limit. At the localization transition $\gamma = 2$ the form of $R(t)$ is scale-invariant. The numerical data shown in the paper is averaged over $N_r = 1000$ disorder realizations and $N_x = 16$ basis states $|\Psi_0\rangle$, $f = 1/8$, $\lambda = 0.1$.

The main results of this paper can be formulated as follows (see also Fig. 1):

- In the ergodic phase $\gamma < 1$ the survival probability with the chosen projector $\widehat{P}_f$, Eq. (4), exhibits pronounced oscillations in time. The amplitude of these oscillations decays as $\propto t^{-2}$ with time and the frequency of the oscillations is given by the spectral width of the initially prepared wave packet and thus is inversely proportional to $f$. The presence of such oscillations and the character of their decay with time in the thermodynamic limit $N \to \infty$ in general depends on the initial spectral decomposition of the prepared wave packet [39–42] and in the case of the projected basis state, Eq. (4) acquires a standard form given by $R(t) \sim [\sin(\pi t / \Delta t) / (\pi t / \Delta t)]^2$, where $\Delta t \propto f^{-1}$. This form is determined only by the Fourier transform of the initial wave packet thus implying an energy-independent overlap of different eigenfunctions valid for fully ergodic states. On the contrary, in the NEE phase mini-bands are formed in the local spectrum which make the eigenfunction overlap essentially energy-dependent resulting in a drastic suppression of oscillations in $R(t)$. Thus the standard projection procedure allows one to use $R(t)$ as a probe for the fully ergodic phase.

- In the NEE phase the survival probability decays exponentially $R(t) = \exp[-\Gamma(N) t]$ with

time as long as $R(t) \gg N^{-D}$. The characteristic decay rate $\Gamma(N) \propto N^{D-1}$ is determined by the fractal dimension $D$ and goes to zero in the thermodynamic limit $N \to \infty$ provided that $D < 1$. Thus the decay rate $R(t)$ in NEE phase is slower than in the ergodic phase albeit it is still exponential. In this phase, some oscillations in $R(t)$ may exist, but only as a finite-size effect.

- In the localized phase $R(t)$ tends to a constant $R(\infty)$ as $t \to \infty$, where $R(\infty)$ remains finite and tends to its initial value $R(0)$ as $N \to \infty$.

## 2  Survival probability in terms of eigenfunctions and eigenvalues

The survival probability $R(t)$ can be expressed in terms of exact eigenvalues $E_n$ and eigenfunctions $\psi_n(r)$ of the state $n$ as:

$$R(t) = \sum_{n,m}' \overline{|\psi_n(0)|^2 |\psi_m(0)|^2 \cos[(E_n - E_m)t]}, \tag{5}$$

where sums run over eigenstates indices, which belong to the projected subspace $\sum_n' = \sum_{n \in (1-f)N/2}^{(1+f)N/2}$ containing a finite fraction $f$ of the system spectrum [3]. Indeed, the initial basis state $\langle r|\Psi_0\rangle \equiv \Psi_0(r, t=0) = \delta_{r,0}$ which is non-zero only on one site $r = 0$, can be represented as follows using the completeness of the set of eigenfunctions:

$$\Psi(r, t = 0) = \sum_n \psi_n^*(0) \psi_n(r). \tag{6}$$

The application of the projector $\widehat{P}_f$ to this state to form the initial wave function $\Psi_f(r, t = 0) \equiv \langle r|\widehat{P}_f\Psi_0\rangle$ implies restriction in the summation to $\sum_n'$. The Schrödinger dynamics $\psi_n(r, t) = \psi_n e^{i E_n t}$ then leads to:

$$\Psi_f(r, t) = \sum_n' \psi_n^*(0) \psi_n(r) e^{i E_n t}, \tag{7}$$

which immediately results in Eq. (5) for $R(t) = \overline{|\Psi_f(0, t)|^2}$.

## 3  Ansatz for eigenvectors of GRP model

The key point of our analytical consideration is the following *ansatz* for the amplitude of an eigenstate of GRP model, $\psi_n(r)$ at site $r$, suggested first in [35] and implicitly derived for local density of states (DoS) from non-linear sigma model in [36] (see also [32, 43]). This ansatz works for all three phases:

$$|\psi_n(r)|^2 = \frac{|H_{nr}|^2}{(E_n - \varepsilon_r)^2 + \Gamma(N)^2}, \tag{8}$$

where $E_n$ and $\varepsilon_n$ are exact and bare energy levels of the wavefunction which differ from each other by the value of the order $\Gamma(N)$.

For NEE states ($1 < \gamma < 2$) [24] the characteristic scale

$$\Gamma(N) = \pi \lambda^2 N^{-\gamma}/\delta = \pi \lambda^2 \rho_0 N^{1-\gamma} \propto \delta N^D, \tag{9}$$

---

[3]The choice of the initial site $r = 0$ is irrelevant after taking disorder average.

has a meaning of the width of the mini-band in the local spectrum, where $\delta = (\rho_0 N)^{-1}$ is the global mean level spacing, and $\rho_0$ is the mean density of states at $E = 0$.

For ergodic states $\gamma < 1$ the global DoS has a semi-ellipse form [24, 44]:

$$\rho(E) = \frac{\sqrt{2S - E^2}}{\pi S}, \quad S = \sum_n \overline{|H_{mn}|^2} = \lambda^2 N^{1-\gamma}, \tag{10}$$

so that in the center of the band $\delta(N) \propto N^{-(1+\gamma)/2}$ [24] and

$$\Gamma(N) = \delta(N) N \propto N^{(1-\gamma)/2} \tag{11}$$

grows with $N$ like the global band-width $\sim N\delta$. It follows from Eq. (8) in this case that the typical and the maximal amplitudes are of the same order $|\psi_n(r)|^2 \sim |H_{nr}|^2 / [\Gamma(N)]^2 \sim N^{-1}$.

For localized states ($\gamma > 2$),

$$\Gamma(N) = \lambda N^{-\gamma/2} \ll \delta \tag{12}$$

is just the mean splitting $\left(\overline{|H_{nm}|^2}\right)^{1/2}$ of two resonance levels [24]. In this case the peak amplitude at $n = r$ is

$$|\psi_n(n)|^2 = 1 + O(N^{-(\gamma-2)}), \tag{13}$$

the maximal $|\psi_n(r)|^2$ of a typical wave function at $r \neq n$ according to Eq. (8) is of the order of

$$|\psi_n(r)|^2_{max} = |H_{nr}|^2 / [\delta^2 + \Gamma^2] \sim |H_{nr}|^2 / \delta^2 \sim N^{-(\gamma-2)} \ll 1, \tag{14}$$

while the typical

$$|\psi_n(r)|^2_{typ} = \exp[\overline{\ln|\psi|^2}] \sim |H_{nr}|^2 / [1 + \Gamma^2] \sim |H_{nr}|^2 \sim N^{-\gamma} \tag{15}$$

scales in the same way with $\gamma$ as in NEE phase [24]. Equation (13) expresses an important property of GRP model that the wave function is essentially localized on one single site, i.e. the localization radius is equal to zero.

## 4 The mini-bands

In the NEE phase, the parameter $\Gamma(N) \sim N^{D-1}$ has the meaning of the width of a *mini-band* [17, 45] in the local spectrum (e.g., in the local DoS) formed by $N^D$ states sharing the same fractal support set in the coordinate space (see Fig. 2). On this set of sites the eigenfunction coefficients of states belonging to a mini-band are large $\sim N^{-D} \gg \langle |\psi|^2 \rangle = N^{-1}$ compared to the average value, while outside of it they are much smaller and correspond to the typical value $\sim N^{-2+D}$ (see Fig. 2).

These 'high amplitude' states are separated by the typical level spacing $\delta_{mb}$ of the order of the global one $\delta \sim N^{-1}$. The entire coordinate space consists of $N^{1-D}$ non-intersecting fractal support sets each one corresponding to a mini-band in the global spectrum (see Fig. 7 in Ref. [45]). Thus the entire global spectrum is a unification of $N^{1-D}$ mini-bands of which typically only one is seen in the local spectrum.

This picture is encoded in Eq. (8). Indeed, the typical energy difference $|E_n - \varepsilon_r| \sim 1$ according to Eq. (8) corresponds to the typical amplitude $|\psi_n(r)|^2_{typ} \sim |H_{nm}|^2 \propto N^{-\gamma} = N^{-2+D}$, while the amplitude of states inside the mini-band $|E_n - \varepsilon_r| < \Gamma(N)$ is much larger $|\psi_n(r)|^2 \sim |H_{nm}|^2 / [\Gamma(N)]^2 \propto N^{-(2-\gamma)} = N^{-D} \gg N^{-2+D}$.

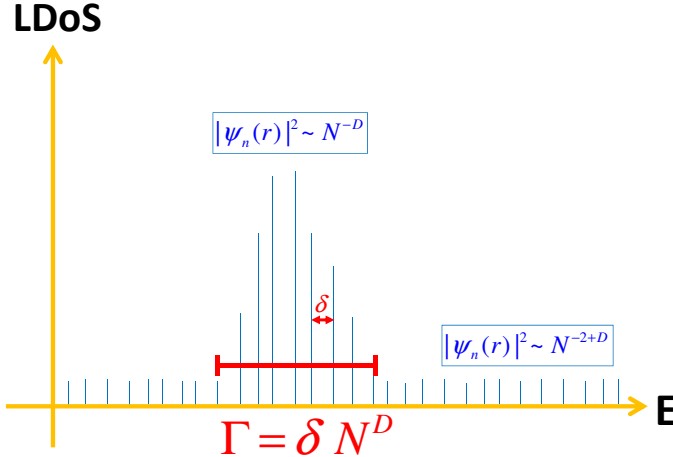

Figure 2: (Color online) A mini-band in the local density of states: $\sim N^D$ states belonging to a mini-band have an amplitude $\sim N^{-D}$ in the observation point $r$ while the rest of the states have a small amplitude $\sim N^{-2+D}$ in this point. The parameter $\Gamma$ in Eq. (8) has a meaning of the typical width of a mini-band.

Furthermore, the picture of Fig. 2 allows to reproduce the analytical results of Ref. [34] on the dependence of the typical and the maximal local DoS (LDoS) on the bare level width $\eta \gg \delta$. The typical value of LDoS,

$$\rho(E, r) = \frac{1}{\pi} \sum_n |\psi_n(r)|^2 \frac{\eta}{(E_n - E)^2 + \eta^2}, \tag{16}$$

corresponds to the position of the observation energy $E$ outside the mini-band. Then the contribution to LDoS from the levels within the energy window $\eta$ centered at the observation energy is the product of the wave function amplitude, the typical value of Lorenzian and the typical number of terms in the sum Eq. (16), respectively: $N^{-2+D} \times \eta^{-1} \times \eta/\delta \sim N^{D-1} \sim \Gamma$. The contribution from the mini-band at a typical distance $\sim 1$ from the observation energy is $N^{-D} \times \eta \times \Gamma/\delta \sim \eta$. Thus we readily obtain the result of Ref. [34]:

$$\rho_{\text{typ}} \sim \begin{cases} \Gamma \sim N^{D-1} & \text{if } \eta < \Gamma \\ \eta & \text{otherwise} \end{cases}. \tag{17}$$

The maximal LDoS corresponds to the observation energy inside a mini-band. Then the similar estimation gives:

$$\rho_{max} \sim \begin{cases} \Gamma^{-1} & \text{if } \eta < \Gamma \\ \eta^{-1} & \text{otherwise} \end{cases}. \tag{18}$$

The averaged LDoS remains independent of $\eta$ value and is of order 1: $\langle \rho \rangle \sim 1$. Note that the distribution of LDoS is highly asymmetric (like in the Anderson insulator) with $\rho_{typ} \ll \langle \rho \rangle \ll \rho_{max}$ but it has a non-singular (like in the metal) limit as $\eta \to 0$. The difference with the metal is that this limit is $N$-dependent.

We believe that the picture of mini-bands (though it is not necessarily a property of multi-fractality) is a general property of slow-dynamic state in many-body systems. The peculiarity of the considered toy model is that the local energy spectrum inside a mini-band is similar to the one in the metal. This results in a simple exponential decay of survival probability obtained below. However, the characteristic decay rate $\Gamma$ tends to zero in the thermodynamic limit $N \to \infty$ which is a signature of slow dynamics in this particular case. Generically, the states inside a mini-band may have a wide (power-law) distribution of local spacings [17] and

form a Cantor set. In this case the survival probability should decay slower than an exponential (presumably streched-exponentially [35, 56]).

We conclude, therefore, that interacting systems may and should show the diversity of behavior that does not literary reduce to the behavior of the RP toy model. However, the classification of this behavior in the slow-dynamics phase should be based on the classification of the types of mini-bands, the simplest of which is represented by the RP random matrix theory (RMT).

# 5 The eigenvector overlap function $K(\omega)$.

An important measure of the wave function statistics is the overlap function [45, 46]

$$K(E_n - E_m) = N \sum_r \overline{|\psi_n(r)|^2 |\psi_m(r)|^2}_{\text{off}}, , \tag{19}$$

it enters matrix elements of all the local interactions. In particular, it is responsible for the enhancement of superconducting transition temperature in dirty metals close to metal-insulator transition [47, 48] and recently it has been suggested to result in an enhancement of phonon relaxation and electron-phonon cooling rates close to the localization transition [49]. Furthermore, $K(\omega)$ is the Fourier-transform of the survival probability which is an important dynamical measure relevant also for many-body localization [42, 50]. Here $\overline{A}_{\text{off}}$ denotes the average of $A$ over the matrix elements $H_{nm}$ keeping the exact energies $E_n$ fixed. [4] Plugging Eq. (8) into Eq. (19), replacing the summation over $r$ by an integration over $\rho_0 \varepsilon_r$ and using the fact that the convolution of two Cauchy functions is also the Cauchy function with double width, one obtains for the delocalized phase in the limit $\Gamma(N) \gg \delta(N)$:

$$K(E_n - E_m) = \frac{1}{\pi \rho_0} \frac{2\Gamma(N)}{(E_n - E_m)^2 + 4\Gamma^2(N)}, \quad (n \neq m), \tag{20a}$$

$$K(0) = N I_2 \equiv N \sum_r \overline{|\psi_n(r)|^4} \sim N^{1-D}. \tag{20b}$$

In the localized phase, where $\Gamma(N) \ll \delta(N)$, the main contribution to Eq. (19) is done by the two terms with $r = n$ and $r = m$. Taking into account that $|\psi_n(n)|^2 \approx 1$, one obtains:

$$K(E_n - E_m) = \frac{2N^{1-\gamma}}{(E_n - E_m)^2 + \Gamma^2(N)}, \tag{21a}$$

$$K(0) = \frac{N^{1-\gamma}}{\Gamma^2(N)} \sim N. \tag{21b}$$

# 6 $R(t)$ and the eigenvalue distribution

Finally we obtain the expression for $R(t)$ in terms of $K(E_n - E_m)$:

$$R(t) = f I_2 + \frac{1}{N^2} \sum_{\substack{n,m \\ n \neq m}}' \overline{K(E_n - E_m) \cos[(E_n - E_m) t]}_{\text{E}}, \tag{22}$$

where $\overline{A}_E$ denotes the average of $A$ over the eigenvalue distribution. In this equation $f$ is the fraction of all the states involved in the projection procedure in Eq. (4).

---

[4]In the thermodynamic limit the energies $E_n$ are statistically independent of *any single* $H_{i\neq j}$, since the difference $E_n - \varepsilon_n$ is contributed by *all* $N^2 - N$ off-diagonal matrix elements of the same order (see also the arguments of self-averaging over off-diagonal matrix elements in [34, 36]).

Eq. (22) can be rewritten

$$R(t) = f I_2 + \int dE \, dE' \, C(E, E') K(E - E') \cos[(E - E') t], \qquad (23)$$

using the DoS correlation function:

$$C(E, E') = \overline{\rho(E) \rho(E')} - N^{-1} \bar{\rho}(E) \delta(E - E'), \quad \rho(E) = \frac{1}{N} \sum_n{}' \delta(E - E_n). \qquad (24)$$

Note that the projected DoS $\rho(E)$ depends on the shape of the initial wave packet which brings about the dependence of $R(t)$ on the initial form of the particle density distribution. In order to separate the effect of initial distribution from that the eigenvalue and eigenfunction statistics encoded in Eq. (8) it is convenient to represent $C(E, E')$ in the following form:

$$C(E, E') = \bar{\rho}(E) \bar{\rho}(E') [1 + G(E - E')]. \qquad (25)$$

One can first integrate over the sum $s = (E + E')/2$ in Eq. (23) and then integrate over the difference $\omega = (E - E')$ with the result:

$$R(t) = f I_2 + \int_{-\infty}^{\infty} d\omega \, F(\omega) (1 + G(\omega)) K(\omega) \cos(\omega t), \qquad (26a)$$

$$F(\omega) = \int_{-\infty}^{\infty} ds \, \bar{\rho}(s + \omega/2) \bar{\rho}(s - \omega/2). \qquad (26b)$$

The three functions $F(\omega)$, $G(\omega)$ and $K(\omega)$ in the integrand of Eq. (26a) describe three different effects on $R(t)$. The eigenvector overlap function $K(\omega)$, Eqs. (20, 21), describes the effect of eigenfunction statistics. The spectral correlation function $G(\omega)$ describes the correlation hole in the level statistics due to the repulsion of energy levels. Its Fourier transform $S(u) - 1$ is given by Eq. (17) and Fig. 7 in Ref. [24]. The 'instrumental' function $F(\omega)$ depends on the shape of the initial wave packet. For the box-shaped $\bar{\rho}(E) = \rho_0 \, \theta(f/2\rho_0 - |E|)$ resulting from the projection with a hard cutoff onto a small fraction $f \ll 1$ of states near $E = 0$ one obtains $F(\omega) = \rho_0^2 \, \theta(f/\rho_0 - |\omega|)(f/\rho_0 - |\omega|)$. This function is non-analytic with the jump of the first derivative both at $\omega = 0$ and at $\omega = f/\rho_0$. This non-analyticity results in a power law term $\propto t^{-2}$ at large $t$, as well as the oscillating term $\propto t^{-2} \cos[2\pi(f/\rho_0) t]$. In contrast, the 'soft' cutoff (e.g. the Gaussian initial wave packet $\bar{\rho}(E) = \rho_0^2 \exp\left[-4\rho_0^2 E^2/f^2\right]$) results in the smooth (e.g.Gaussian) $F(\omega)$ which does not give rise to the power-law and oscillating terms in $R(t)$. We will see, however, that the projection with the hard cutoff is very useful for characterization of phases by the behavior of the oscillating term.

## 7    $R(t)$ at $t < t_E$.

The Fourier transform $\tilde{G}(t)$ of $G(\omega)$ tends to zero at $t \gg t_H = 1/\delta$. However, for $1 < \gamma \leq 2$ it is exactly zero also at $t = 0$ [24]. This means that there is a time and thus an energy scale where $\tilde{G}(t)$ takes a minimal value. According to Eq. (17) of Ref. [24] this scale, which has a meaning of the Ehrenfest time [51] for the present problem, is given by:

$$t_E = E_{Ehr}^{-1} = \frac{1}{2} \Gamma^{-1} \ln(N^D) \sim N^{1-D} \ln(N^D) \ll t_H. \qquad (27)$$

The fact that $\tilde{G}(t = 0) = 0$ implies that the positive area under the plot of $G(\omega)$ at large $|\omega|$ is exactly equal to the negative area at small $|\omega|$ (see Fig. 3). The latter is of order $\delta$. This gives

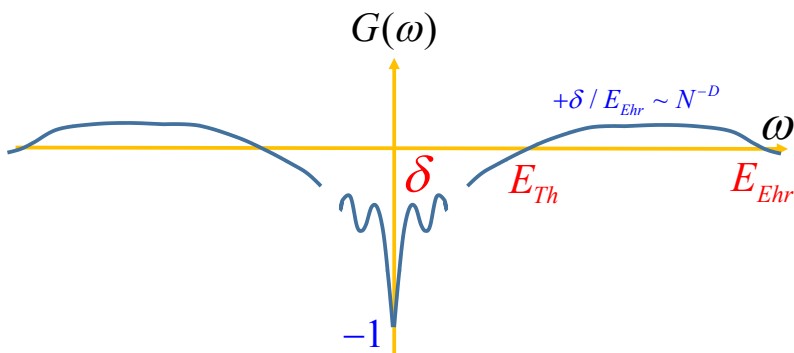

Figure 3: (Color online) Sketch of the function $G(\omega)$ for $1 < \gamma < 2$. The positive and negative area under the plot are exactly equal to each other due to $\tilde{G}(t=0) = 0$. Level repulsion implies $G(\omega = 0) = -1$. For small frequencies $\omega \lesssim E_{Th}$ the behavior is GOE. At intermediate frequencies $\sqrt{\delta E_{Ehr}} \sim E_{Th} \lesssim \omega \lesssim E_{Ehr} \sim N^{1-\gamma}$ the function $G(\omega)$ changes sign and is of the order of $\delta/E_{Ehr} \sim N^{-D}$ [52].

$G(\omega) \sim \delta/E_{Ehr}$ in a wide interval $\omega \lesssim E_{Ehr}$ down to the Thouless energy $E_{Th} \ll E_{Ehr}$ below which $G(\omega)$ behaves as in GOE. In particular for $\delta \lesssim \omega \lesssim E_{Th}$ the behavior is $G(\omega) \sim -(\delta/\omega)^2$ [38]. From the matching at $\omega \sim E_{Th}$ one obtains $E_{Th} \sim \sqrt{E_{Ehr}\delta}$ which with the accuracy up to a $\ln N$ factor coincides with the result of Ref. [52] [5].

As $G(\omega)$ decreases fast at $\omega \gg E_{Ehr}$ it is safe to neglect $G(\omega)$ in Eq. (26a) at times $t \lesssim t_E$. We also adopt a hard cutoff projection procedure which corresponds to the instrumental function and its Fourier transform

$$F(\omega) = \rho_0^2 \, \theta(f/\rho_0 - |\omega|)(f/\rho_0 - |\omega|); \quad \tilde{F}(t) = 4\rho_0^2 \frac{\sin^2\left(\frac{f t}{2\rho_0}\right)}{t^2}. \tag{28}$$

Then Eq. (26a) reduces to the convolution:

$$R(t) = f \, I_2 + 2\pi \rho_0^{-1} \int_{-\infty}^{+\infty} \tilde{F}(t') e^{-2\Gamma|t-t'|} \, dt'. \tag{29}$$

It is convenient to introduce the dimensionless parameters:

$$c = \frac{f}{2\rho_0\Gamma(N)}, \quad k = t\,\Gamma(N)/\pi. \tag{30}$$

Then in extended phases $0 < \gamma < 2$ one obtains:

$$R(t) = f \, I_2 + f \, Y\left(\frac{t\Gamma}{\pi}, c\right), \tag{31}$$

where in the limit $2\pi k \sqrt{c+1} \gg 1$ the function $Y(k,c)$ is very well approximated (see Fig. 4) by:

$$Y(k,c) \approx e^{-2\pi|k|} + \frac{1}{2\pi c (\pi k)^2} - \frac{\cos(2\pi kc)}{2\pi c (\pi k)^2 (c^2+1)}. \tag{32}$$

Thus we conclude from Eqs. (31, 32) that $R(t)$ has an oscillating part which period of oscillations is:

$$\Delta t = \frac{2\pi \rho_0}{f}. \tag{33}$$

---

[5]Note that in Ref. [24] there is a different definition of the Thouless energy and time. Here we stick to the original definition of the Thouless energy given in [53].

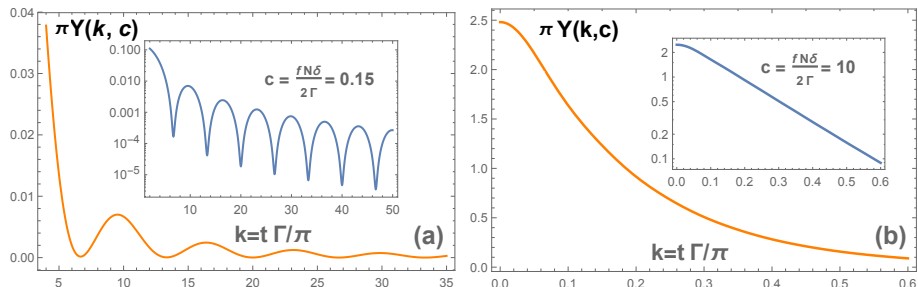

Figure 4: (Color online) $Y(k,c)$ (a) from Eq. (32) for $c = 0.15$ and (b) from Eq. (32) for $c = 10$. Insets show the same function in log-linear scale. In the ergodic phase $c \simeq f \ll 1$ is $N$-independent and small at small $f$. In the non-ergodic extended and localized phases $c \propto N^{\gamma-1}$ and $c \propto N^{\gamma/2}$, respectively. It increases with increasing $N$, and oscillations disappear in the thermodynamic limit.

This period is $N$-independent for $\gamma > 1$ and for $\gamma < 1$ it is proportional to $N^{-(1-\gamma)/2}$.

For ergodic phase, $\gamma < 1$, in agreement with Eqs. (32) at $c \ll 1$ and Eq. (31) the survival probability $R(t)$

$$R(t) - R(\infty) = R(0) \left[ \frac{\sin(\pi t/\Delta t)}{\pi t/\Delta t} \right]^2 \tag{34}$$

coincides with the Fourier transform (28) of the box-shaped DoS of states involved in the projection procedure in Eq. (4) (see Fig. 4(a)). These oscillations are *standard* for fully ergodic phases, behaving as Gaussian ensembles [38] and have the same origin as those considered in [39–41]. The difference is that in the above references *all* states participate in the initial wave packet, so that the decay of oscillations $\sim t^{-3}$ is determined by the branch singularity of the semi-circular mean DoS at the band edge and not by the hard cutoff projection procedure which results in the decay $\sim t^{-2}$. [6]

For NEE phase, $1 < \gamma < 2$, corresponding to large $c \sim \Gamma^{-1}(N) \sim N^{\gamma-1}$ in Eq. (32), oscillations are suppressed in the thermodynamic limit (see Fig. 4(b)). This suppression can be traced back to the presence of mini-bands in the local spectrum which width, $\Gamma$, is vanishing in the thermodynamic limit. Therefore, we believe that the suppression of oscillations is a generic property of the NEE phase.

Note that the coefficient in front of $Y(k,c)$ in Eq. (31) is an $N$-independent constant for extended states ($0 < \gamma < 2$) and it is proportional to $N^{1-\gamma/2}$ for localized states ($\gamma > 2$). In the latter case $R(t) \equiv R(t \to 0)$ in the limit $N \to \infty$.

## 8  $R(t)$ at $t > t_E$.

For large times $t \gg t_E$ (or small $\omega \ll E_{Ehr}$) the 'instrumental' function $F(\omega)$ can be replaced by a constant. Furthermore, the exponentially decaying term $e^{-2\Gamma t}$ which results from the first term in $(1 + G(\omega))$ in Eq. (26a) also does not play an essential role. In this case $R(t)$ can be approximated by:

$$R(t) = f I_2 + 2\pi f \int_{-\infty}^{\infty} \tilde{G}(t - t') e^{-2\Gamma |t'|} dt' \approx f I_2 + 2\pi f \Gamma^{-1} \tilde{G}(t), \tag{35}$$

where the Fourier transform of $G(\omega)$, $\tilde{G}(t) = (\mathscr{R}(t) - 1)\delta$, is related to the spectral form-factor $\mathscr{R}(t)$ computed in Ref. [24].

---

[6]The $t^{-2}$ decay caused by the hard cutoff projection is reminiscent of the $t^{-2}$ decay of oscillations in realistic lattice many-body quantum systems [39–41]

Thus we conclude that the large-t behavior of the return probability is entirely determined by the spectral correlations [39–41]. Furthermore, $\mathscr{R}(t)$ has a minimum at $t \sim t_E$ and increases almost linearly $\mathscr{R}(t) - 1 \approx t/t_H - 1$ in the interval $E_{Th}^{-1} = t_{Th} \lesssim t \lesssim t_H \sim \delta^{-1}$ to reach the saturation value $\mathscr{R}_\infty = 1$ at $t > t_H$ (see Fig. 7 of Ref. [24]). This linear behavior stems from the corresponding behavior of the Gaussian RMT and implies the 'overshooting' in $R(t)$ seen in numerical simulations shown in the inset of Fig. 8.

# 9 The spectral rigidity

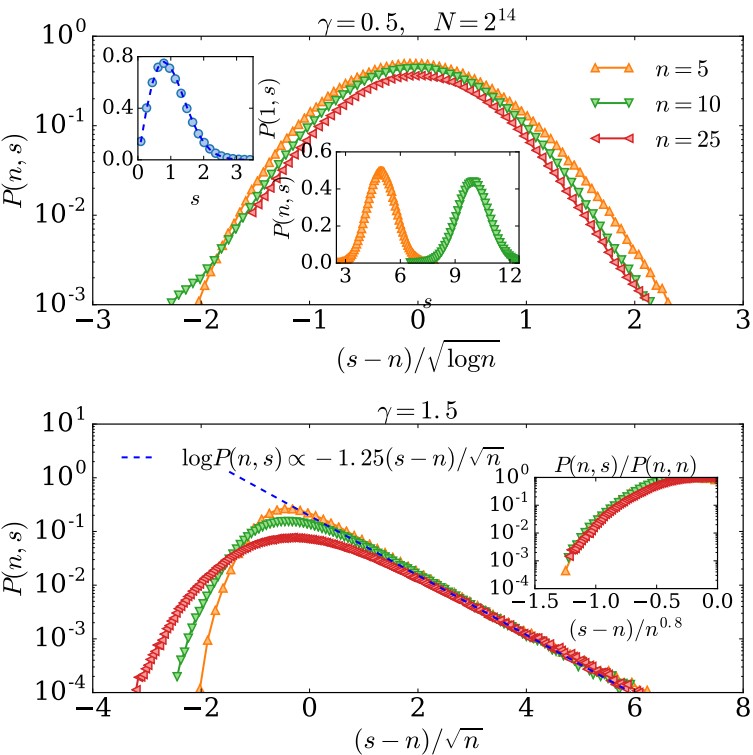

Figure 5: (Upper panel): Gaussian $\ln P(s, n) \sim -(s-n)^2/\mathrm{var}(n)$ in the ergodic phase $\gamma = 0.5$. The variance $\mathrm{var}(n) \sim \ln(n)$ as it should be for the Wigner-Dyson statistics. (lower panel): $P(s, n)$ in the NEE phase $\gamma = 1.5$. The behavior of $P(s, n)$ for $s > n$ is quasi-Poisson with the variance $\mathrm{var}_R(n) = \chi n$, and level compressibility $0 < \chi < 1$. In contrast, for $s < n$ the function $\ln P(s, n)$ is non-linear in $s$ with the variance $\mathrm{var}_L(n) \sim n^{1.6}$. Such a super-Poisson behavior reflects the mini-band structure of spectrum with level correlations inside mini-bands much stronger than between of them.

For random Hamiltonians the spectrum is also random but exhibits the Wigner-Dyson *spectral rigidity* in the *ergodic phase*. It implies that the variance $\mathrm{var}(n)$ of the number of levels in an energy interval containing $n$ levels on average,

$$\mathrm{var}_{\mathrm{erg}}(n) \propto \ln(n), \quad n \gg 1, \tag{36}$$

is small compared to statistically independent (Poissonian) fluctuation of levels, typical for localized states:

$$\mathrm{var}_{\mathrm{loc}}(n) = n. \tag{37}$$

The level rigidity has been suggested long time ago as the test for ergodicity, criticality and localization [53–55] . Indeed critical states at the Anderson transition point have the quasi-Poisson level number variance:

$$\text{var}_{\text{cr}}(n) = \chi\, n, \quad (0 < \chi < 1). \tag{38}$$

A related quantity is the probability density $P(s, n)$ to have energy difference $\Delta E = s\,\delta$ be-

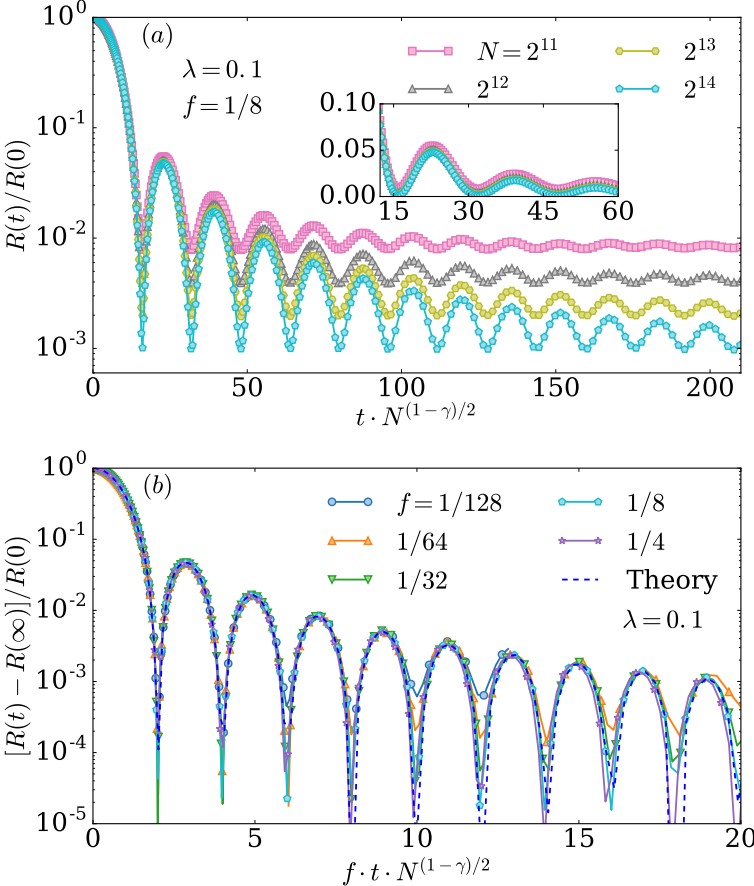

Figure 6: (Color online) Oscillations of survival probability $R(t)$ in the ergodic phase, $\gamma = 0.25$ (a) for different matrix sizes $N$ and (b) for different fraction of states $f$ involved. The period of oscillations scales as $N^{-(1-\gamma)/2}$, in accordance with Eq. (33). In the ergodic phase $c \ll 1$ the form of oscillations is well approximated (see Eqs. (31, 32)) by Eq. (34) shown as the dashed blue line.

tween two levels, provided that in between of them there are exactly $n-1$ other levels. Thus $P(s, 1) = P(s)$ is the well-known spacing distribution function, decaying exponentially for the Poisson level statistics $P(s) = e^{-s}$ and showing the level repulsion $P(0) = 0$ for the Wigner-Dyson ensembles $P(s) = (\pi s/2)e^{-\pi s^2/4}$. In Fig. 5 we plot the results obtained by exact diagonalization of GRP model for $P(s, n)$ statistics.

The upper panel demonstrates that in the ergodic phase $P(s, n)$ has a Gaussian form $\ln P(s, n) = -(s-n)^2/\text{var}(n)$, where $\text{var}(n) \sim \ln(n)$, as it should be for the Wigner-Dyson level statistics.

In the NEE phase the character of $P(s, n)$ changes drastically. First, $P(s, n)$ for $s > n$ has a quasi Poisson tail $\ln P(s, n) \sim -(s-n)/\text{var}_R(n)$ with $\text{var}_R(n) = \chi\, n$, $(0 < \chi < 1)$ as in Eq. (38). However, for $n \gg 1$ it makes sense also to study $P(s, n)$ for $s < n$ and $|s-n| \ll n$ (lower-panel

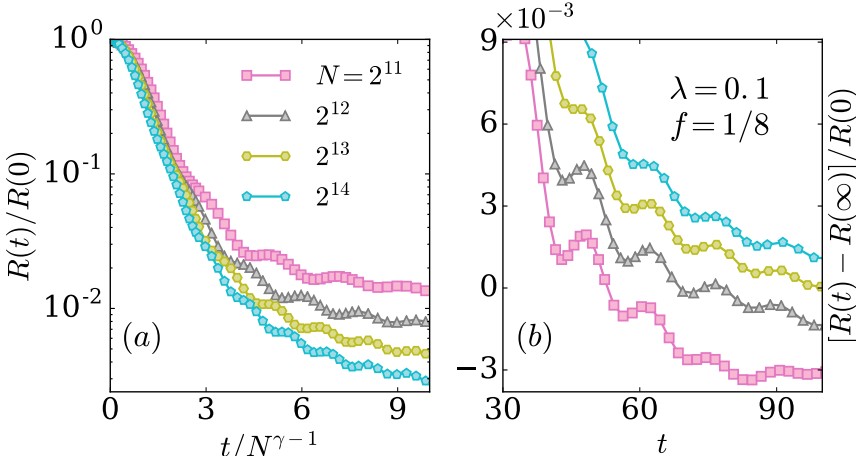

Figure 7: (Color online) Residual oscillations in $R(t)$ in the NEE phase with $\gamma = 1.25$ and $\lambda = 1$ versus (a) rescaled time collapsing exponential decay, (b) unscaled time. Oscillations decay with the increasing system size and the exponential behavior Eq. (32) emerges. Period of oscillations does not scale with the system size in accordance with Eq. (33).

in Fig. 5). A remarkable fact is that the curves for $P(s, n)$ for $s < n$ and different $n$ collapse with the variance $\text{var}_L(n) \sim n^\beta$, $\beta = 1.6 > 1$, for $\gamma = 1.5$. Such a *super-Poisson* behavior of the variance implies clustering which is related to the mini-band structure [17] where the level correlations inside a cluster of levels forming a mini-band are much stronger than between the mini-bands.

## 10 Oscillations in $R(t)$ as a numerical test for ergodicity

Figure 6 shows $R(t)$ obtained by numerical diagonalization of GRP matrix Hamiltonians of the size $N$ up to $N = 2^{14}$ [7] in the ergodic phase, $\gamma = 0.25$. The results, which should be compared with Fig. 4(a), are *quantitatively* described by Eq. 34, see the theoretical dashed curve in Fig. 6. The visibility of oscillations in $R(t)$ does not depend on the system size.

As has been shown above analytically, in the NEE phase the oscillations in $R(t)$ is a finite-size effect controlled by the parameter $c \sim \Gamma^{-1}(N)$. They are present at a finite $N$ close to ergodic transition at $\gamma = 1$ when $c \propto \lambda^{-2} N^{\gamma-1}$ is not large enough and can be suppressed either by increasing the system size or by decreasing a value of the constant $\lambda$ in Eq. (2), cf. Figs. 7 and 8.

In Fig. 7 we demonstrate how with increasing the system size in the NEE phase, $\gamma > 1$, the oscillations die out giving way to the exponential decay. We also demonstrate that the exponential decay has a different scaling with $N$ than the oscillating part, the period of oscillations being $N$-independent.

Thus we conclude that the behavior of oscillations in GRP model with increasing the system size can serve as a test for ergodicity. Their visibility drops dramatically when the mini-bands are formed in the local spectrum signaling of the lack of ergodicity. We believe that this test is useful not only in GRP but also in other models [39–41,56], including many-body quantum

---

[7]The numerical data shown in the paper is averaged over $N_r = 1000$ disorder realizations and $N_x = 16$ basis states $|\Psi_0\rangle$.

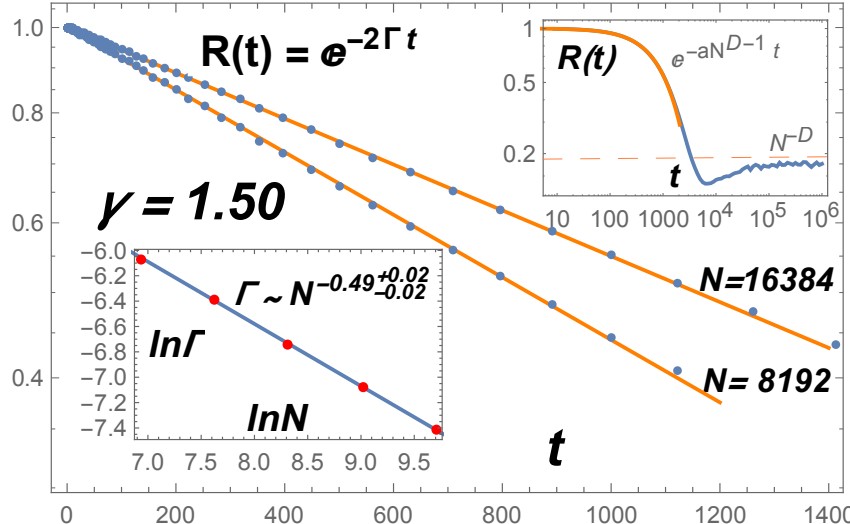

Figure 8: (Color online) Exponential decay of survival probability in NEE phase of GRP model with $\gamma = 1.5$, $\lambda = 0.1$. (upper inset) Global view of survival probability including the asymptotic time-independent regime. $R(t)$ reaches the asymptotic value $\sim N^{-D}$ from below showing the 'overshooting'. (lower inset) Extraction of $D \approx 0.49 \pm 0.02$ at $\gamma = 1.5$ from the decay rate $2\Gamma(N) \propto N^{-1+D}$.

systems [8].

## 11  Extracting fractal dimension $D$ from $R(t)$ in the NEE phase

As it follows from Eq. (30) the parameter $c \propto N^{1-D}$ in Eq. (32) determines $R(t)$ via Eq. (31), and it is infinite in the thermodynamic limit $N \to \infty$ in the NEE phase since $D < 1$. In this case $R(t) = e^{-2\Gamma(N)t}$ at large enough $N$. Then the scaling of the mini-band width, Eq. (9), gives access to the fractal dimension $D$ through measuring the decrement $2\Gamma(N)$ of the exponential decay. Figure 8 (which should be compared with Fig. 4(b)) shows an example of extraction of $D(\gamma)$ at $\gamma = 1.5$ from the finite time dynamic of $R(t)$, which has been obtained using exact diagonalization.

Figure 9 demonstrates that this method of extraction of $D$ from the scaling with $N$ of the decay rate $\Gamma(N) \propto N^{-\zeta}$, Eq. (9), is very accurate, and it does not suffer from large finite-size corrections. It should be noted that the alternative method of extraction of $D$ from the asymptotic value $R(\infty)$ of $R(t)$ (shown by a dashed line in upper inset in Fig. 8) requires much larger system sizes and thus much more computational efforts.

## 12  $R(t)$ in the localized phase

In Fig. 1 we give an overview of the evolution of the form of $R(t)$ with increasing $\gamma$ in all three phases. The localization transition at $\gamma = 2$ is marked by the scale-independent $R(t)$ that has a finite limit $R(\infty) \propto I_2$ at $N \to \infty$. In the localized phase $\gamma > 2$ the limiting value $R(\infty)$ remains finite.

For finite $N$ the asymptotic value $R(\infty)$ depends on the constant $\lambda$ in Eq. (2), as shown in

---

[8]However, in some numerical works (see, e.g., [42]), the oscillations of the survival probability in a many-body interacting spin system survive even in the localized phase.

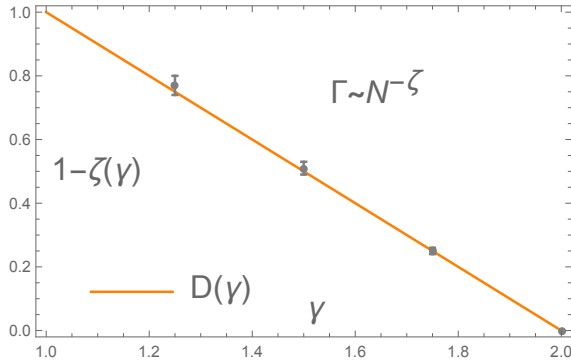

Figure 9: (Color online) Fractal dimension $D(\gamma) = 1 - \zeta$ extracted numerically from $\Gamma(N) \propto N^{-\zeta}$ using Eq. (9) and the theoretical prediction $D = 2 - \gamma$ of Ref. [24] (solid line).

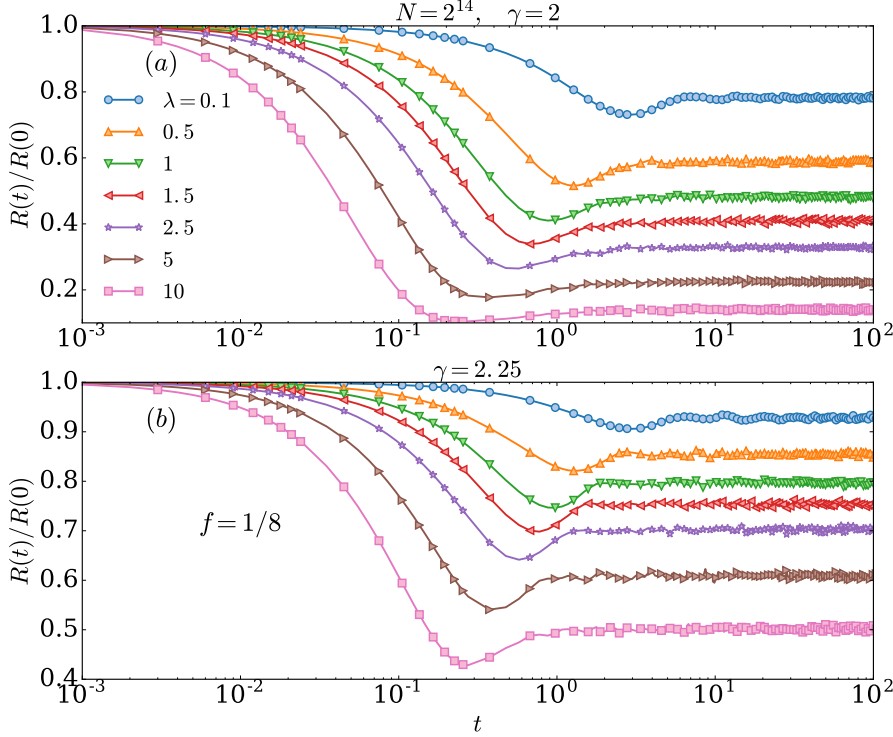

Figure 10: (Color online) Finite-$N$ survival probability $R(t)$ (a) at the critical point of localization transition $\gamma_c = 2$ and (b) in the insulating phase $\gamma = 2.25$ for different values of $\lambda$ in Eq. (2). The parameters are $N = 2^{14}$ and $f = 1/8$.

Fig. 10. At a fixed $\lambda$ and increasing $N$ the value $R(\infty)$ stays constant in the critical point $\gamma_c = 2$ of the localization transition but it moves towards its initial value $R(0)$ in the insulating phase (see Fig. 11 and Fig. 12). This is related with the special property [24] of the localized phase in GRP ensemble that there is a gap between the peak value, Eq. (13), of $|\psi_n(n)|^2 \approx 1$ and the typical maximal value of $|\psi_n(r \neq n)|^2 \sim N^{-(\gamma-2)} \ll 1$ (see Eq. (8) and [32] for details). This implies that the Lyapunov exponent is divergent in the thermodynamic limit, as for the Bethe lattice with infinite connectivity [57]

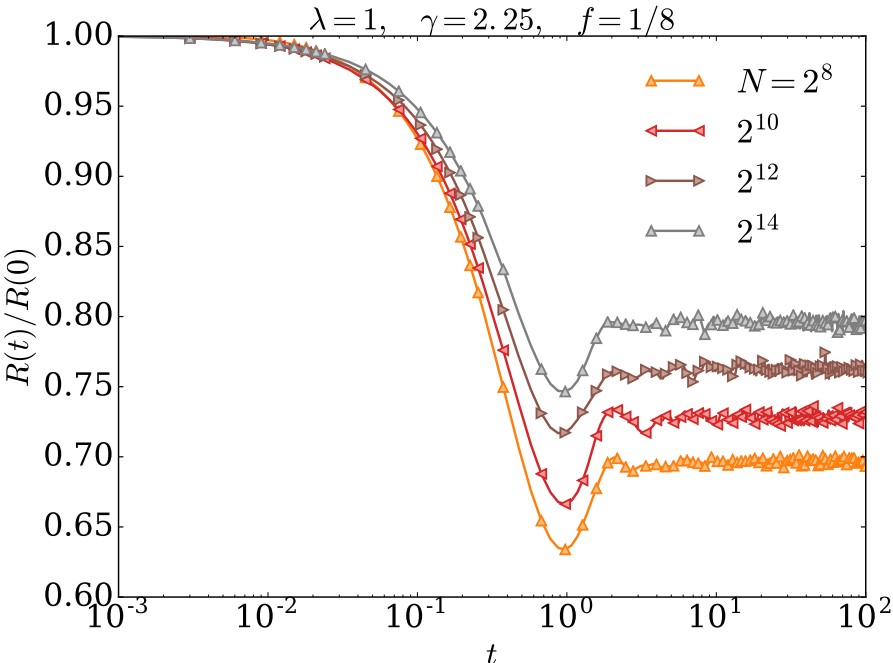

Figure 11: (Color online) Evolution of $R(t)$ in the insulating phase at $\gamma = 2.25$ with $\lambda = 1$ and $f = 1/8$. With increasing $N$ the asymptotic value $R(\infty)$ moves towards its initial value $R(0)$.

## 13   Conclusions and Discussion

In summary, the detailed analysis of the survival probability $R(t)$ defined in Eq. (4) in the generalized Rosenzweig-Porter ensemble [24] demonstrates three distinctly different types of behavior in different phases. The ergodic phase shows robust oscillations in $R(t)$ with a standard polynomial decay $\sim t^{-2}$ of their amplitude with time and a universal frequency equal to the bandwidth of the initially prepared wavepacket. These oscillations essentially depend on the initial spectral decomposition of the prepared wave packet and in the case of the projected basis state, Eq. (4), provide a probe of ergodicity of the extended states. In the multifractal phase no oscillations persist in the thermodynamic limit and the survival probability decays exponentially, albeit with a small rate which is related to the fractal dimension of the eigenstates. An exponential decay of the survival probability in this phase is nearly free of the finite size effects and provides an accurate way to extract the multifractal dimension of eigenstates from the decay rate. Localized phase is characterized by the universal size-independent saturation value $R(t \to \infty) = R(\infty)$ coinciding with the initial value $R(t \to 0) = R(0)$ in the thermodynamic limit $N \to \infty$.

We have shown that several characteristic dynamical features are caused by the formation of *mini-bands* in the NEE phase. Among them: (i) small and vanishing in the thermodynamic limit decrement of the exponential decay of $R(t)$; (ii) the decreasing and eventually vanishing visibility of oscillations in $R(t)$ with increasing the matrix size; (iii) super-Poissonian behavior of $P(s, n)$ at $0 < s < n$. While the point (i) explicitly implies slow dynamics in our model, we believe that the effects (ii) and (iii) should also be relevant in many-body systems in their slow dynamics regime.

A more subtle issue which goes beyond the scope of this paper is the relation between multifractality and sub-diffusion. This issue is important, as sub-diffusion is the most frequently discussed feature of the slow dynamics regime. We know [58] that in the one-particle local-

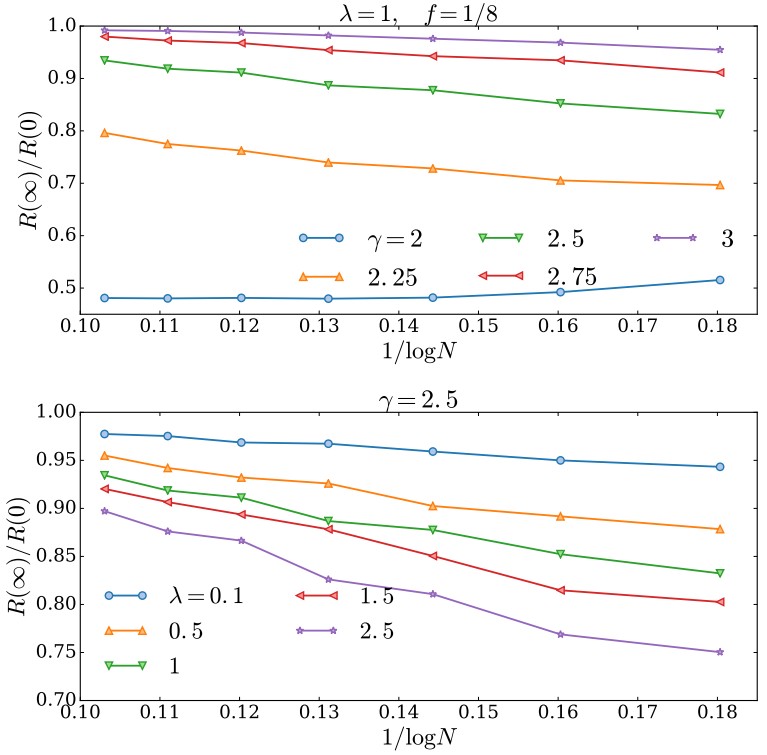

Figure 12: (Color online) The ratio $R(\infty)/R(0)$ as a function of $1/\ln N$ (upper) for different values of $\gamma$ at $\lambda = 1$ and $f = 1/8$ and (lower) for different values of $\lambda$ at $\gamma = 2.5$. For all $\gamma > 2$ in the insulating phase $R(\infty)/R(0)$ increases with increasing the system size and should approach 1 in the thermodynamic limit. At the localization transition $\gamma_c = 2$ the ratio $R(\infty)/R(0)$ is saturated for $N > 10000$ at a $\lambda$-dependent value smaller than 1.

ization problem sub-diffusion $\langle \mathbf{r}^2 \rangle \propto t^{2/d}$ in the critical point of the Anderson transition on $d > 2$ dimensional lattices follows from the *one-parameter scaling* which does not necessarily assume multifractality. On the other hand, for the critical states in the center of Landau band in the integer quantum Hall effect in two-dimensional systems, multifractality is present but the sub-diffusion is not [59]. Therefore, multifractality do not necessarily mean sub-diffusion and vice versa. Note also that multifractality does not imply existence of mini-bands either, as in the well studied analytically example of Critical Power-Law Banded Random Matrices (CPLBRM) [60] mini-bands are absent.

The straightforward attack to compute $\langle \mathbf{r}^2 \rangle \equiv \langle (n-r)^2 \rangle$ as a function of $t$ when the wave packet is initially created at a site $n$, done in Ref. [61] for GRP and CPLBRM resulted in the diffusion spreading but with divergent diffusion coefficient $\sim N^{3-\gamma}$ and $\sim N$, respectively. This divergence is expected for systems with long-range hopping. The problem with the RMT as toy models for sub-diffusion in many-body systems is that in such systems diffusion or sub-diffusion occurs in the real space, while the RMT models eigenfunctions in the Hilbert space which is exponentially larger. Therefore, in order to make the RMT toy models like GRP or CPLBRM useful for describing sub-diffusion in many-body systems one should find a proxy for the real space and project the RMT eigenfunctions onto this subspace. So far this program has not been done.

A very recent piece of evidence of the relevance of GRP model for real systems and computer science came from the preprint [25]. It has been shown in this paper that the simplest

model for a quantum glass transition, the Quantum Random Energy Model, is reduced to GRP-like model and shows the characteristic features of the spin autocorrelation function similar to the ones of return probability $R(t)$ studied in this paper.

# 14 Acknowledgements

We are grateful to B. L. Altshuler, L. B. Ioffe and M. V. Feigel'man for stimulating discussions on the applicability of this model to Quantum Random Energy Problem. IMK acknowledges the support of the Russian Foundation for Basic Research Grant No. 17-52-12044 and German Research Foundation (DFG) Grant No. KH 425/1-1. SB acknowledges support from DST, India, through Ramanujan Fellowship Grant No. SB/S2/RJN-128/2016. VEK and SB acknowledge hospitality of the Max-Planck Institute for the Physics of Complex Systems, Dresden, Germany. MA acknowledges hospitality of the International Centre for Theoretical Physics, Trieste, Italy. VEK acknowledges hospitality of KITP at UCSB where the final part of the work was done and the support from the National Science Foundation under Grant No. NSF PHY-1748958.

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
