# Peer review of "Survival probability in Generalized Rosenzweig-Porter random matrix ensemble"

_SciPost Physics, doi:SciPost Phys. 6, 014 (2019)_

## Round 2 · Referee Report · Anonymous (Referee 1) · 2018-11-18

Referee report on the resubmitted paper "Survival probability in Generalized
Rosenzweig-Porter random matrix ensemble."

by G. De Tomasi, M. Amini, S. Bera , I. M. Khaymovich and V. E. Kravtsov

The paper contains a detailed analysis of decay of a prepared wavepacket whose
dynamics is driven by a random matrix Hamiltonian of (generalized) Porter-Rozenzweig
model. The model attracted considerable attention in recent years as, possibly, the
symplest nontrivial toy model for a system with non-ergodic multifractal eigenstates.
Depending on the control parameter $\gamma$ eigenstates also can be fully extended (a.k.a.
ergodic) or localized. This gives an opportunity to probe all regimes in a unified
one-parameter framework.

The main ingredient of the analysis is the *Ansatz* Eq.(8) which postulates that in
all three phases the eigenstates profile is Lorentzian, with the effective widths $\Gamma(N)$
reflecting absence or presence of ergodicity by its dependence on the system size
$N$. The ensuing analysis of the wavepacket dynamics is relatively straightforward,
though I would like to praise the authors for their accurate and careful description of
the procedure and illuminating discussion of various dynamical regimes. The main
practical message is that it is much more reliable to extract the multifrcatal expo-
nents from the exponential decay of survival probability in the multifrcatal phase
rather than from eigenfunction moments. This is an interesting observation, and
may be of essential practical utility. Altogether, the paper is certainly a useful ad-
dition to the growing literature on Porter-Rozenzweig model, and I can recommend
it for publication, after the authors take into account the following remark.

My main concern is that when discussing the main Ansatz Eq.(8) the authors

- (**i**) only very grudgingly mention the paper [21] by C. Monthus. As far as
  I can see, it would be more fair to explicitly state that the Ansatz was first
  proposed in that paper, which arrived to it by semi-heuristic Wigner-Weisskopf
  type arguments, and only then to give references to much more recent [28] and
  to yet unpublished [18].

- (**ii**) completely disregard the the paper by A. Ossipov & K. Truong (EPL 116
  (2016) 37002) which contains results highly relevant to this formula.

In fact I would dare to say that in certain sense that work is presently the best
analytical justification of the above Ansatz. Indeed, it was shown there (although

not in the most explicit form) that the Local Density of States for the same model is given precisely by Lorentzian form with the same width $\Gamma(N)$, and further the expression for $\Gamma(N)$ is derived in the multifractal phase explicitly, with all due prefactors.

Indeed, Eq.(7) in A. Ossipov & K. Truong for the moments of the eigenvectors contains the Lorentzian (and for $q = 1$ it gives essentially an expression for the LDOS). The width of the Lorentzian, denoted there as $s$, can be found from Eq.(11) of that paper in the case of the random diagonal part (relevant to the Rosenzweig-Porter model). In that case, it was shown that the parameter $s$ becomes self-averaging and can be replaced by its mean $\langle s \rangle$.

When the variance of the diagonal matrix elements (denoted *sigma* in that paper), becomes large, then $\langle s \rangle$ can be calculated asymptotically. The corresponding result is given by their Eq.(17). For the Rosenzweig-Porter model, $\sigma^2 = N^{\gamma-1}$, which is large for $\gamma > 1$. Substituting $\sigma = N^{(\gamma-1)/2}$ into Eq.(17), one obtains for $E/\sigma << 1$

$$\langle s \rangle = \sqrt{\pi/2} N^{-(\gamma-1)/2}.$$

which precisely gives an asymptotic expression for the width of the Lorentzian (including a prefactor!). This expression was actually used further in that work to derive Eqs.(20)-(21) for the eigenfunction moments.

To summarize, the work by A. Ossipov & K. Truong contains valuable information corroborating with the main Ansatz and must be added to the list of references and included into the discussion. The paper by C. Monthus ought to be given full credit for introducing the Ansatz itself.

---

## Round 2 · Referee Report · Anonymous (Referee 2) · 2018-11-19

Strengths

1- very interesting

2- very clear

Weaknesses

no weakness

Report

The GRP model is known as the simplest matrix model where
a Non-Ergodic Extended phase with multifractal properties
exists besides the Localized phase and the Ergodic phase.
Here the authors study the dynamical properties of the model,
and obtain that the survival probability R(t) can be used as a clear signature
to distinguish the three phases.
The various regimes and approximations are clearly explained,
with well chosen figures displaying the corresponding numerical data.
This work is definitely very interesting within the field of random matrix models,
but also relevant for the field of Many-Body-Localization as explained in the Introduction.

As a consequence, I strongly recommend the publication of this manuscript.

Requested changes

no change are required

---

## Round 2 · Referee Report · Anonymous (Referee 3) · 2018-12-6

Strengths

1- synergy of analytical and numerical approaches
2- very careful numerical analysis
3- simple interpretation of numerical results in analytical terms
4- very high expertise of the authors

Weaknesses

1- relevance to MBL seems superficial in the present version of the manuscript; the authors should elaborate on this point (see report)
2- some definitions in Sec. 3 should be made more transparent
3- some numerical observations require more intuitive explanations

Report

The manuscript addresses numerically and analytically dynamics in the generalized Rosenzweig-Potter model (GRPM) which is a toy-model for studying non-ergodic multifractal phase. The time evolution of the survival probability $R(t)$ defined through a certain projection on a small fraction of eigenstates was investigated for ergodic, multifractal, and localized phases, and the fractal exponent was extracted from the behavior of $R(t)$. The paper presents a very thorough numerical analysis of the model, which is very timely and important in view of the hot debates in the community on the existence of delocalized non-ergodic phases in physical models. The analytical consideration based on the simple Ansatz (18) is by and large consistent with the numerical observations.

Before recommending this interesting work for publication, I would like to clarify the following points:

1. As a major motivation for this study, the authors mention the possibility of a subdiffusive behavior in the delocalized phase in the problem of many-body localization (MBL). I certainly agree with the authors that this is indeed an important motivation, but it is not immediately clear to me whether the analysis of the GRPM sheds more light on that issue. First, it would be nice to be sure that subdiffusion requires non-ergodicity and non-ergodicity always manifests itself in many-body problems through a subdiffusive behavior. Second, in contrast to the GRPM (where the existence of nontrivial exponents is kind of put in by hand in the formulation of the problem -- through the statistics of matrix elements), the subdiffusive behavior in the conventional setting for the MBL problem (with short-range interactions) is an emergent phenomenon. Therefore, I believe that it would be very beneficial for the readers if the authors elaborated on the potential relevance of their results to MBL in the discussion section. Currently, the connection between the two problems appears to be rather superficial, in my view. A a side remark, I would like to propose the authors to add some recent reference in connection to the subdiffusive behavior in the MBL problem, e.g., arXiv:1603.06588, arXiv:1807.05051, arXiv:1809.02137, arXiv:1809.02894, and, perhaps, some other works that contributed to the debate on whether subdiffusion is a transient or a true thermodynamic phenomenon. I also think that it would be useful to mention earlier works where the survival probability was studied in a many-body setting in connection to the relaxation in quantum dots (Altshuler-Gefen-Kamenev-Levitov model), in particular,
P.G. Silvestrov, Phys. Rev. B 64, 113309 (2001).

2. The observed behavior of $R(t)$ seems to contradict the above-mentioned potential relation between the GRPM and the subdiffusive behavior in interacting models. Indeed, it turns out that $R(t)$ in the ergodic phase of the GRPM decays slower (as $1/t^2$) than in the multifractal phase (where the decay is exponential). For a subdiffusive behavior, one would naively expect the opposite -- because of the increased return probability characteristic of subdiffusion. The counter-intuitive faster decay in the non-ergodic phase requires a simple explanation.

3. It is not quite clear to me whether the $1/t^2$ decay of $R(t)$ in the ergodic phase is not an artifact of the choice of the initial state which is produced by projecting on the window of eigenstates with hard boundaries. If instead of the hard cutoff in the definition of $\hat{P}_f$ one uses a soft window of the same width determined by $f$ (this can be achieved by introducing decaying coefficients $c_n$ in the definition of $\hat{P}_f$), would not this eliminate the peculiar features observed for the hard window? The argument behind this is quite simple: the Fourier transform of a box is different from the Fourier transform of, say, a Gaussian. Of course, the study with the chosen form of $\hat{P}_f$ is interesting and instructive on its own; the question is rather about universality of the findings in the manuscript.

4. Definitions in Sec. 3 require clarification. Indeed, there two types of labels in the problem: $r$ for the sites, and $n$ for the eigenenergies. Quite unexpectedly, Eq. (8) uses the site label $r$ for the energy $\varepsilon_r$. The relation only briefly mentioned above Eq. (8) and is rather confusing (... with its maximum occuring at $r=n$). What is $\varepsilon_r$? As far as I see, the formal "algorithm" is as follows: (i) label sites by $r$; (ii) solve for for the eigenenergies and eigenfunctions -- this gives an order set of $\varepsilon_n$ with the corresponding $\psi_n(r)$; (iii) for each $n$ starting with $n=1$ find $r_m(n)$ such that $|\psi_n(r_m)|^2$ is maximum out of all $|\psi_n(r)|^2$; (iv) based on this relabel the site $r_m(n)=n$ everywhere; (v) repeat for the next $n$. This should completely relabel all the sites compared to the original labeling used for finding the eigenenergies. I believe this should be clearly explained. In this regard, while such relabeling is natural for localized states, where the eigenfunctions are dominated by single site and hence there is a clear correspondence between eigenenergy index and the site label, it is rather tricky to associate some single site to a given eigenstate $\psi_n$ -- this is the reason for my confusion.

In conclusion, the manuscript can be reconsidered for publication after the authors have considered the above points. I strongly believe that the authors would find my remarks, questions, and suggestions useful for further improving their paper.

Requested changes

1- update references to the subdiffusion in the many-body setting
2- further elaborate on the relevance of the GRPM to MBL
3- discuss universality of the results with respect to the form of $\hat{P}_f$
4- clarify definitions and labeling used in Sec. 3

(see report for details)

---

## Round 3 · Author Response

Dear Editor,

We thank all the referees for the careful reading of our paper and for useful comments.
Hereby we resubmit our manuscript to SciPost Physics.
Please see below the list of changes in the manuscript and our reply to the comments below.
List of main changes:

Sincerely yours,
Giuseppe De Tomasi, Mohsen Amini, Soumya Bera , Ivan M. Khaymovich and Vladimir E. Kravtsov.
* * *
============================================
Anonymous Report 1 on 2018-11-18
============================================
The paper contains a detailed analysis of decay of a prepared wavepacket whose dynamics is driven by a random matrix Hamiltonian of (generalized) Porter-Rozenzweig model. The model attracted considerable attention in recent years as, possibly, the symplest nontrivial toy model for a system with non-ergodic multifractal eigenstates. Depending on the control parameter γ eigenstates also can be fully extended (a.k.a. ergodic) or localized. This gives an opportunity to probe all regimes in a unified one-parameter framework.
The main ingredient of the analysis is the Ansatz Eq.(8) which postulates that in all three phases the eigenstates profile is Lorentzian, with the effective widths Γ(N ) reflecting absence or presence of ergodicity by its dependence on the system size N . The ensuing analysis of the wavepacket dynamics is relatively straightforward, though I would like to praise the authors for their accurate and careful description of the procedure and illuminating discussion of various dynamical regimes. The main practical message is that it is much more reliable to extract the multifrcatal exponents from the exponential decay of survival probability in the multifrcatal phase rather than from eigenfunction moments. This is an interesting observation, and may be of essential practical utility. Altogether, the paper is certainly a useful addition to the growing literature on Porter-Rozenzweig model.

Reply
We thank the Referee for high evaluation of our work.
* * *
I can recommend it for publication, after the authors take into account the following remark.
My main concern is that when discussing the main Ansatz Eq.(8) the authors
• (i) only very grudgingly mention the paper [21] by C. Monthus. As far as I can see, it would be more fair to explicitly state that the Ansatz was first proposed in that paper, which arrived to it by semi-heuristic Wigner-Weisskopf type arguments, and only then to give references to much more recent [28] and to yet unpublished [18].
• (ii) completely disregard the the paper by A. Ossipov & K. Truong (EPL 116 (2016) 37002) which contains results highly relevant to this formula.

In fact I would dare to say that in certain sense that work is presently the best analytical justification of the above Ansatz. Indeed, it was shown there (although not in the most explicit form) that the Local Density of States for the same model is given precisely by Lorentzian form with the same width Γ(N ), and further the expression for Γ(N ) is derived in the multifractal phase explicitly, with all due prefactors.
Indeed, Eq.(7) in A. Ossipov & K. Truong for the moments of the eigenvectors contains the Lorentzian (and for q = 1 it gives essentially an expression for the LDOS). The width of the Lorentzian, denoted there as s, can be found from Eq.(11) of that paper in the case of the random diagonal part (relevant to the Rosenzweig- Porter model). In that case, it was shown that the parameter s becomes self- averaging and can be replaced by its mean (s).
When the variance of the diagonal matrix elements (denoted sigma in that pa- per), becomes large, then (s) can be calculated asymptotically. The corresponding result is given by their Eq.(17). For the Rosenzweig-Porter model, σ2 = N γ−1, which is large for γ > 1. Substituting σ = N (γ−1)/2 into Eq.(17), one obtains for E/σ << 1
(s) = /π/2N −(γ−1)/2.
which precisely gives an asymptotic expression for the width of the Lorentzian (including a prefactor!). This expression was actually used further in that work to derive Eqs.(20)-(21) for the eigenfunction moments.
To summarize, the work by A. Ossipov & K. Truong contains valuable informa- tion corroborating with the main Ansatz and must be added to the list of references and included into the discussion. The paper by C. Monthus ought to be given full credit for introducing the Ansatz itself.

Reply
We thank the referee for pointing out both relevant papers by Monthus and by Truong and Ossipov.
It is our fault that the paper by Truong and Ossipov was absent in the first version of the paper.
Now we have added the references to Truong and Ossipov and gave a full credit to both works.

============================================
Anonymous Report 2 on 2018-11-19
============================================
Strengths
1- very interesting
2- very clear

Weaknesses
no weakness

Report
The GRP model is known as the simplest matrix model where a Non-Ergodic Extended phase with multifractal properties exists besides the Localized phase and the Ergodic phase.
Here the authors study the dynamical properties of the model, and obtain that the survival probability R(t) can be used as a clear signature to distinguish the three phases.
The various regimes and approximations are clearly explained, with well chosen figures displaying the corresponding numerical data.
This work is definitely very interesting within the field of random matrix models, but also relevant for the field of Many-Body-Localization as explained in the Introduction. As a consequence, I strongly recommend the publication of this manuscript.

Requested changes
no change are required

Reply
We thank the Referee for high evaluation of our work and results and for accepting our paper in the current form for the publication.

============================================
Anonymous Report 3 on 2018-12-6
============================================
Strengths
1- synergy of analytical and numerical approaches
2- very careful numerical analysis
3- simple interpretation of numerical results in analytical terms
4- very high expertise of the authors
Weaknesses
1- relevance to MBL seems superficial in the present version of the manuscript; the authors should elaborate on this point (see report)
2- some definitions in Sec. 3 should be made more transparent
3- some numerical observations require more intuitive explanations
Report

The manuscript addresses numerically and analytically dynamics in the generalized Rosenzweig-Potter model (GRPM) which is a toy-model for studying non-ergodic multifractal phase. The time evolution of the survival probability R(t) defined through a certain projection on a small fraction of eigenstates was investigated for ergodic, multifractal, and localized phases, and the fractal exponent was extracted from the behavior of R(t). The paper presents a very thorough numerical analysis of the model, which is very timely and important in view of the hot debates in the community on the existence of delocalized non-ergodic phases in physical models. The analytical consideration based on the simple Ansatz (18) is by and large consistent with the numerical observations.

Reply
We thank the Referee for high evaluation of our work.
* * *
Before recommending this interesting work for publication, I would like to clarify the following points:
1. As a major motivation for this study, the authors mention the possibility of a subdiffusive behavior in the delocalized phase in the problem of many-body localization (MBL). I certainly agree with the authors that this is indeed an important motivation, but it is not immediately clear to me whether the analysis of the GRPM sheds more light on that issue. First, it would be nice to be sure that subdiffusion requires non-ergodicity and non-ergodicity always manifests itself in many-body problems through a subdiffusive behavior. Second, in contrast to the GRPM (where the existence of nontrivial exponents is kind of put in by hand in the formulation of the problem – through the statistics of matrix elements), the subdiffusive behavior in the conventional setting for the MBL problem (with short-range interactions) is an emergent phenomenon. Therefore, I believe that it would be very beneficial for the readers if the authors elaborated on the potential relevance of their results to MBL in the discussion section. Currently, the connection between the two problems appears to be rather superficial, in my view.

Reply
We agree that the issue of sub-diffusion and its relation with multifractality is not properly discussed in the paper. A straightforward attack on this problem in GRP ensemble (see, e.g., the work [M. Amini, Europhys. Lett. 117, 30003 (2017)] on the wavepacket spreading in this model) results in a singular diffusion with divergent diffusion coefficient. However, it is not relevant for the many-body systems in slow dynamics regime, as diffusion in many-body (MB) systems happens in the real space, while the eigenvectors of GRP ensemble are analogous to eigenfunctions in the Hilbert space. Since the correspondence between the real space and the exponentially larger Hilbert space is subtle due to different notion of locality, any non-local in the real space problem such as the wavepacket spreading requires the proxy of the real space in the RMT models. This problem is not solved so far.
We added the corresponding discussion in the Conclusion.

However, slow dynamics regime in MB problems does not reduce to merely sub-diffusion. It also implies slow dynamics in the survival probability, a local property which proxy in the real space, namely, the spin autocorrelation function is known. We have shown that in GRP model the slowness of dynamics of survival probability is due to the presence of mini-bands in the local spectrum which width is vanishing in the thermodynamic limit. We believe that the presence of mini-bands (possibly with a Cantor set structure) is also a characteristic feature of the MB systems in the slow dynamic phase, and this is why the GRP is useful as a toy model for MB systems.
The recent piece of evidence of the relevance of GRP model for quantum spin glass and for computer science came with the preprint arXiv:1812.06016 where it is shown that the simplest model of quantum spin glass, the Quantum Random Energy Model (QREM), reduces to a kind of GRP random matrix theory and that the spin autocorrelation function in QREM essentially coincides with the return probability R(t) in GRP ensemble.
* * *
As a side remark, I would like to propose the authors to add some recent reference in connection to the subdiffusive behavior in the MBL problem, e.g., arXiv:1603.06588, arXiv:1807.05051, arXiv:1809.02137, arXiv:1809.02894, and, perhaps, some other works that contributed to the debate on whether subdiffusion is a transient or a true thermodynamic phenomenon. I also think that it would be useful to mention earlier works where the survival probability was studied in a many-body setting in connection to the relaxation in quantum dots (Altshuler-Gefen-Kamenev-Levitov model), in particular,
P.G. Silvestrov, Phys. Rev. B 64, 113309 (2001).

Reply
We have added the references suggested by the Referee.
* * *
2. The observed behavior of R(t) seems to contradict the above-mentioned potential relation between the GRPM and the subdiffusive behavior in interacting models. Indeed, it turns out that R(t) in the ergodic phase of the GRPM decays slower (as 1/t2) than in the multifractal phase (where the decay is exponential). For a subdiffusive behavior, one would naively expect the opposite -- because of the increased return probability characteristic of subdiffusion. The counter-intuitive faster decay in the non-ergodic phase requires a simple explanation.

Reply
This comment seems to be a result of confusion. Indeed, both the prefactor in front of the $1/t^2$ decay and the exponential decay rate $\Gamma(N)$ depend on the system size N and are small in the fractal and the localized phases. As in the ergodic phase the prefactor in front of the $1/t^2$ decay is N-independent, the power-law decay of $R(t)$ in the ergodic state is always faster than the exponential decay with the small N-dependent decrement $\Gamma(N)\sim N^{-1+D}$ in the fractal phase (see, e.g., Fig. 1 in the manuscript where the time is rescaled with $\Gamma(N)$ in the fractal phase).
* * *
3. It is not quite clear to me whether the 1/t2 decay of R(t) in the ergodic phase is not an artifact of the choice of the initial state which is produced by projecting on the window of eigenstates with hard boundaries. If instead of the hard cutoff in the definition of P^f one uses a soft window of the same width determined by f (this can be achieved by introducing decaying coefficients cn in the definition of P^f ), would not this eliminate the peculiar features observed for the hard window? The argument behind this is quite simple: the Fourier transform of a box is different from the Fourier transform of, say, a Gaussian. Of course, the study with the chosen form of P^f is interesting and instructive on its own; the question is rather about universality of the findings in the manuscript.

Reply
We agree with the Referee that the presence of oscillations in R(t) is due to the projection procedure that we adopted. However, this very projection may serve as an instrument to distinguish the ergodic phase (where the visibility of oscillations does not depend on the matrix size) and the multifractal phase (where oscillations die out in the thermodynamic limit).

We emphasized this point in the revised version. Moreover, we presented a new analytical derivation of R(t) which explicitly separates the effect of the initial condition, the eigenfunction correlations and the eigenvalue correlations.
* * *
4. Definitions in Sec. 3 require clarification. Indeed, there two types of labels in the problem: r for the sites, and n for the eigenenergies. Quite unexpectedly, Eq. (8) uses the site label r for the energy εr. The relation only briefly mentioned above Eq. (8) and is rather confusing (... with its maximum occurring at r = n). What is εr? As far as I see, the formal "algorithm" is as follows: (i) label sites by r; (ii) solve for the eigenenergies and eigenfunctions -- this gives an order set of εn with the corresponding ψn(r); (iii) for each n starting with n = 1 find rm(n) such that|ψn(rm)| is maximum out of all |ψn(r)| ; (iv) based on this relabel the site rm(n) = n everywhere; (v) repeat for the next n. This should completely relabel all the sites compared to the original labeling used for finding the eigenenergies. I believe this should be clearly explained. In this regard, while such relabeling is natural for localized states, where the eigenfunctions are dominated by single site and hence there is a clear correspondence between eigenenergy index and the site label, it is rather tricky to associate some single site to a given eigenstate ψn -- this is the reason for my confusion.

Reply
We thank the Referee for pointing out this issue. As a matter of fact it was caused by a misprint. One of the energies in this ansatz should be the true eigenenergy of the state, whereas the other is the diagonal matrix element $H_{rr}$ in the point of observation. We corrected the misprint and emphasized this point to avoid further confusion.
* * *
In conclusion, the manuscript can be reconsidered for publication after the authors have considered the above points. I strongly believe that the authors would find my remarks, questions, and suggestions useful for further improving their paper.

Requested changes

1- update references to the subdiffusion in the many-body setting
2- 2- further elaborate on the relevance of the GRPM to MBL
3- discuss universality of the results with respect to the form of P^f
4- clarify definitions and labeling used in Sec.3

Reply
We thank the Referee once again for a really highly professional Report of an expert in the field. We have taken into account all the comments of the Referee and now we hope that he/she will agree to publish our manuscript.

---

## Round 3 · List of Changes

List of main changes:
1. The discussion of the connection between MBL models and generalized Rosenzweig-Porter ensemble (GRP) has been extended. A new section concerning the mini-bands in the local spectrum of GRP and many-body systems in the slow dynamics regime has been added. The discussion about the relation between multifractality, sub-diffusion and mini-bands has been added in the Conclusion.
2. The references suggested by the Referees have been added to the reference list.
3. Some other relevant references have been also added.
4. In Section 3 the full respect has been given to the works by Monthus and by Truong and Ossipov which contributions are significant to the derivation of the used ansatz for the wavefunction.
5. The new, more transparent derivation of the form of R(t) is presented both for small and large times and three different effects on R(t) are identified and analyzed in detail. Such derivation does not need Supplementary Material any more.
6. We have emphasized more the point about the chosen projector allowing to use the return probability R(t) as a dynamical indicator of the ergodic phase in random-matrix and MBL models.
7. We have corrected the misprint in the ansatz for eigenfunction amplitude which lead to a confusion of the Referee.

Invited Reports on this Submission

Toggle invited reports view

Anonymous Report 1 on 2019-1-8

Report

In the resubmitted version the authors incorporated all major suggestions from the referees,
which made their presentation both more clear and more fair. I no longer see the points for immediate
and obvious improvement.

  • validity: high
  • significance: high
  • originality: good
  • clarity: high
  • formatting: perfect
  • grammar: good

Contributed Reports on this Submission

Show/hide contributed reports

Anonymous Report 2 on 2019-1-8

Strengths

see previous report

Weaknesses

see present report

Report

The authors have done a good job addressing the points raised by the two referees.

The only point that requires modification concerns the sentence in parenthesis above Eq. (1):
"see, e.g., the discussion of R(t) [12] in the MBL model suggested in the seminal MBL work [1]."
This sentence implies the existence of a time machine, as Ref. [12] was published before Ref. [1].
The authors apparently had in mind another seminal work:
B. L. Altshuler, Y. Gefen, A. Kamenev, and L. S. Levitov, Phys. Rev. Lett. 78, 2803 (1997),
which laid foundation to the modern developments in the field of MBL by discovering a mapping between the interacting disordered many-body systems and non-interacting Anderson localization on certain graphs -- localization in Fock space (which, in fact, nicely resonates with the motivation in the present study).

With this change the manuscript can be published.

Optional: The authors might find it useful to comment on the relation between their findings and those in a very recent preprint, arXiv:1812.10283, where a very similar wording was used in connection to a many-body system: "We report fully ergodic eigenstates in the delocalized phase (irrespective of the computational basis), while the MBL regime features a generically (basis-dependent) multifractal behavior, delocalized but non-ergodic."

Requested changes

1 - change sentence about Ref. [12]
2- add reference to Phys. Rev. Lett. 78, 2803 (1997)

  • validity: high
  • significance: high
  • originality: high
  • clarity: high
  • formatting: perfect
  • grammar: excellent

---

## Round 4 · Author Response

Dear Editor,

We are grateful to all the Referees for their comments and efforts in order to improve our manuscript.
Hereby we resubmit the manuscript with all requested changes.

Sincerely yours,
Giuseppe De Tomasi, Mohsen Amini, Soumya Bera , Ivan M. Khaymovich and Vladimir E. Kravtsov.

---

## Round 4 · List of Changes

1) Page 3: the comment about Ref. [12] has been corrected according to the suggestion of the Referee 2 and the corresponding reference [13] has been added. 2) Page 3: the footnote referring to arXiv:1812.10283 (Ref. [15]) has been added according to the optional suggestion of the Referee 2. 3) The relevant recent references [14, 25-28] have been added with the corresponding comments in pages 3, 18. 4) Some minor language corrections have been included throughout the text.

---

## Editorial Decision

published